# Evolutionary disruption of S&P 500 trading concentration: An intriguing tale of a financial innovation

S. Gowri Shankar [ORCID] *, James M. Miller [ORCID], P. V. (Sundar) Balakrishnan

School of Business, University of Washington, Bothell, Washington, United States of America

* shankar@uw.edu

## Abstract

The novel finding of Balakrishnan, Miller & Shankar (2008) that investors, overwhelmed by the plethora of stock investment offerings, limit their analysis and daily choices to only a small subset of stocks (i.e., herding behavior) now seems to be common wisdom (Iosebashvili, 2019). We investigate whether the introduction of an innovation in financial products designed to allow investors to trade the entire product bundle of S&P 500 stocks, namely S&P 500 index funds, altered "herding behavior" by creating a new class of index investors. We model the distribution of daily trading concentration as a power law function and examine changes over the last six decades. Intriguingly, we discover a unique pattern in the trading concentration distribution that exhibits two distinct trends. For the period 1960–75, the trading concentration of the S&P 500 stocks tracks the increasing trend for the entire market, i.e., the unevenness in trading has steadily increased. However, after the introduction of S&P 500 index funds in 1975, concentration of trading in the S&P 500 stocks has steadily decreased, i.e., trading distribution has become more even across all 500 stocks, contrary to the current belief of equity analysts. This is also in sharp contrast to the case of U.S. stocks that are not in the S&P 500 index where trading concentration has steadily increased. We further corroborate the uniqueness of the inverted V-shape by a counterfactual investigation of the trading concentration patterns for other sets of 500 stock portfolios. This uniquely distinctive trading concentration pattern for S&P 500 stocks appears to be driven by the increasing dominance of bundle trading by index investors.

## 1. Introduction

For the period 1962 to 2005, Balakrishnan, Miller & Shankar [1] modeled the distribution of daily stock trading volumes as a power law function for the NYSE, AMEX and Nasdaq stock markets. They discovered a novel evolutionary phenomenon in which the power law exponents steadily increased over time. Their finding of a non-static, evolutionary trend was striking as it was contrary to most studies in physical science, social sciences, and finance [e.g., 2–4] which report stable exponents over time. Their finding was important and substantive because it indicated that daily trading had become more concentrated in a small set of stocks

**Data Availability Statement:** All data used in this study was retrieved from the Center for Research in Security Prices (CRSP) maintained at the University of Chicago and is available to those with a subscription to this service. Accessing the data is more completely described at www.CRSP.com.

**Funding:** The author(s) received no specific funding for this work.

**Competing interests:** The authors have declared that no competing interests exist.

over time despite an increase in the number of stocks traded between 1962 and 2005. Specifically, during that time period, the number of stocks traded on the US stock markets increased from about 1440 stocks in 1962 to about 6550 in 2005, as recently reported in the Center for Research in Security Prices (CRSP) database. We find the number of stocks traded reached a peak of about 8430 stocks in 1997 but has declined since then. These trends of increases and declines in US stock offerings has been extensively studied in Gao, Ritter, and Zhu [5] and Doidge, Karolyi, and Stultz [6,7].

The result reported above of increasing concentration in stock trading by Balakrishnan, Miller, and Shankar [1] is consistent with the force of unification described by Zipf's Principle of Least Effort [8]. That is, investors, faced with a large set of stock investment offerings to choose from, simplify their selection process and limit their analysis and daily choices in an *a la carte* fashion to only a small subset of stocks. It now seems to be common wisdom that the "huge world of investible assets has shrunk down to a small cohort" [9].

In this paper, we investigate whether the introduction of an innovation in financial products, namely S&P 500 index funds [10], provided investors with an opportunity to alleviate their problem of too many choices. The S&P 500 itself, purely as a measure of market performance, was first proposed in 1957. However, it was only in the mid-1970s that this new product bundling innovation of the S&P 500 index fund was provided to investors. The S&P 500 index funds are purposefully designed to allow investors to buy and sell the entire basket of 500 stocks in the index without analyses of the respective merits of the individual stocks. Thus, the advent of the S&P 500 index funds created a new class of investors, viz., index fund investors who buy and sell entire baskets of stocks.

If this financial innovation does indeed provide investors with a solution to the choice problem, we should expect to observe two special behaviors in the exponents for the distribution of trading in the S&P 500 stocks. First, the pattern of the exponents' behavior should change around the time the S&P 500 index funds as a bundled product were introduced. Before this new product introduction, we expect the exponent to trend upward similar to the trend of the total market (i.e., trading would be more concentrated in a small cohort of stocks). After the introduction, we expect to see a more uniform distribution of trading in all S&P 500 stocks, as index investors trade the entire basket of stocks instead of only a select few stocks. The evolution of the magnitude of the concentration in trading the 500 stocks subsequent to the introduction of the index funds is unknown. This may depend upon the changing influence of the new class of index traders that are being created.

Second, in recent years, the role of index fund investors has grown substantially, with trading in index ETF's (exchange traded funds) alone accounting for over 30% of the total dollar volume of trading in US stock markets [11]. As trading volume in S&P 500 index funds increases over time, we would expect the exponent for the S&P 500 stocks to decrease over time reflecting greater uniformity in the trading of these stocks. This would be reflective of index-fund traders becoming more dominant in the trading of S&P 500 stocks, and would be in stark contrast to the upward trend in the exponent for the total market reflective of greater concentration of trading for the total market.

In this paper, we follow Balakrishnan, Miller & Shankar [1], hereafter BMS [1], in modeling the distribution of daily trading volume (in dollars) as a power law function and using the power law exponents from this distribution as a measure of trading concentration. We refer to these power law exponent values as the Trading Concentration Index or Indices (TCI) throughout this paper. High TCI or power law exponent values indicate that trading is concentrated in a small cohort of stocks, whereas low TCI or power law exponent values suggest a more even distribution of trading across all stocks.

Using this methodology, we find an extremely intriguing empirical result. We discover two distinct trends in the trading concentration index. For the period from 1960 to 1975 the concentration of the daily trading volumes in the 500 stocks that make up the S&P 500 index tracks the entire market; in both cases, the TCI steadily increased. This marks a time period when, although the S&P 500 index existed as a gauge of stock market performance, there were no S&P 500 index funds available to trade the index. After 1975, however, the concentration of trading in the S&P 500 stocks has steadily decreased and the trading distribution has become more even across all the stocks in the index. This trend reversal coincides with the emergence of S&P 500 index fund investors that began in 1976. It appears that the index stocks that were ignored or neglected in the pre-index fund years are now more regularly traded because they are part of the index. In fact, Hegde and McDermott [12] report that the trading volume and institutional investor interest in stocks added to the S&P 500 index increases immediately after the inclusion of the stock in the index, without any other information driving the increase.

The above finding is in sharp contrast to the case of U.S. stocks that are not in the S&P 500 index, as well as for randomly drawn portfolios of 500 stocks. We find that trading concentrations for a portfolio of non-S&P 500 stocks have steadily increased from 1960 to 2018, indicating that trading volumes in these non-S&P500 stock portfolios have increasingly tended to cluster around fewer stocks over time. A counterfactual analysis of randomly drawn portfolios of 500 stocks tracks the TCI for non-S&P 500 stocks over the same period, i.e., the TCI for the trading volumes of these randomly drawn portfolios of 500 stocks also shows an increase in trading concentration. Additionally, an investigation of the entire market of stocks shows a similarly increasing trend in TCI over time. This pattern of an increase in trading concentration for the entire market mimics the finding in the 2008 BMS paper which employed a slightly different measure of trading volume, namely, number of shares traded.

The results for non-S&P 500 stocks, random portfolios, and the entire market, support the widely held belief that large market cap or high visibility stocks attract the most attention from analysts and investors, leaving a large number of smaller stocks to languish as 'neglected' stocks with very little analyst coverage or interest [9]. By one estimate, about 35% of all listed stocks fall in the 'neglected' stocks category, with no analyst coverage whatsoever [13]. Investors in non-index stocks appear to focus on a few, popular stocks resulting in high trading volumes for these stocks; for most other stocks, i.e., 'neglected' stocks, the trading volumes would be relatively light.

This paper makes several contributions. First, we show that the introduction of index funds as a financial innovation in the 1970's marked a sea change in the distribution of trading volumes for stocks included in the index. The increasing popularity of this index fund innovation has led to a more even (or less concentrated) distribution of trading volumes across all S&P 500 stocks over time. Second, we show that for two subsets–the entire stock market excluding the S&P 500 stocks; and randomly constructed portfolios of 500 stocks–trading concentration indices increased over time, similar to the results shown for the entire market in BMS [1]. Third, results shown for the entire market in BMS [1] are independently substantiated by analyzing daily trading dollar volume of all stocks over six decades. Fourth, other insights relating to trading concentration of the largest 500 stocks and the S&P 500 market cap are also presented.

In the next section, we describe the methodology employed, specifically how TCI can be used to measure trading volume concentration. In the third section, we discuss the financial innovation of S&P 500 index funds and our theoretical predictions. In section four, we provide a discussion of our results and the associated implications. In section five, we discuss our conclusions and suggest future research directions.

## 2. Power law exponent as a measure of trading concentration

The power law has been used to model a wide range of economic and financial phenomena. Gabaix [14] provides a comprehensive analysis of the literature in this field. Axtell [2] finds that the distribution of firm sizes in the US is described by Zipf's law [8], a special case of the power law, with a power law exponent of 1.0. Gopikrishnan et al. [15] show that the distribution of trading volume for an individual stock for the time 1994–1995 obeys a power law function with an average exponent of approximately 1.5 for each of the largest 1000 stocks.

Naldi [16] was the first to show that the power law exponent could also be used as a general measure of market concentration, with larger exponent values indicating greater concentration. Traditionally, the Hirschman-Herfindahl Index (HHI) has been used to measure concentration in markets. However, Naldi [16] and Balakrishnan, et al [1] report that the power law exponent is more useful for studying changes in concentration than the HHI measure when dealing with a large number of observations. This is because the huge fluctuations in the daily HHI concentration measure, due to its extreme elasticity, obscure any trends over time. BMS [1] model the distribution of daily trading volumes in the NYSE, AMEX and Nasdaq markets from 1962 to 2005 and find that power law exponents have steadily increased indicating that daily trading has become more concentrated over time. In a subsequent paper, Balakrishnan, Holland, Miller and Shankar [17] also employ the power law exponent as a measure of trading concentration and find it is impacted by market closures and seasonality. Specifically, they report that trading concentrations are lower on the first trading day of the week; on the last two days of the quarter; and on the last few days of the calendar year.

Following Naldi [16] and Balakrishnan, et al. [1], a generalization of the power law to any phenomena whose distribution can be ranked by some metric of size, can be typically expressed as follows:

$$\text{Rank}_i{}^q = \text{constant}/\text{Size}_i \tag{1}$$

In our study of daily trading volumes, we can sort each stock traded on a given day by its trading volume. We can then use the rank order of the volume of each stock $i$ (i.e., labelled as $\text{VolumeRank}_i$) of all stocks traded on any given day, and its corresponding volume traded that day (i.e., Trading Volume$_i$), in operationalizing the power law expression as:

$$\text{VolumeRank}_i{}^q = \text{constant}/\text{Trading Volume}_i \tag{2a}$$

This can be expressed equivalently as:

$$\text{Log}(\text{VolumeRank}_i) = (1/q) * \text{Log}(\text{constant}) - (1/q) * \text{Log}(\text{Volume}_i) \tag{2b}$$

where the power law exponent 'q' for the variable being examined (i.e., trading volume) is our measure of trading concentration. As mentioned earlier, we refer to this power law exponent as the Trading Concentration Index, or TCI, throughout the paper.

Based on the above approach, we model the daily distribution of the trading volumes of individual firms on the stock markets as a power law distribution and estimate the TCI using the formulation:

$$\text{Log}(\text{VolumeRank}_{it}) = \alpha_t - \beta_t * \text{Log}(\text{TradingVolume}_{it}) \tag{3}$$

The Trading Volume is simply the shares traded multiplied by closing price; this is a normalized figure computed as trading volume for a specific firm $i$ divided by average volume across all shares traded on day $t$. The variable VolumeRank$_{it}$ is the rank of the $i$th firm's trading volume on day $t$. We use the ordinary least squares method to estimate the regression coefficients of Eq 3, as recommended by Gabaix [14]. The reciprocal of our estimate of $\beta_t$ is the TCI

for day $t$. A high value for the TCI would suggest that trading is concentrated in a small subset of stocks on day $t$ whereas a low value for the TCI would mean that trading on that day is not concentrated but is spread evenly across all stocks.

## Illustration

| Stock | June 1st | | | October 1st | | |
|---|---|---|---|---|---|---|
| | Rank | Trading Volume | % of total volume | Rank | Trading Volume | % of total volume |
| A | 1 | 150 | 30.0% | 2 | 180 | 36.0% |
| B | 2 | 130 | 26.0% | 1 | 210 | 42.0% |
| C | 3 | 120 | 24.0% | 3 | 70 | 14.0% |
| D | 4 | 100 | 20.0% | 4 | 40 | 8.0% |
| Total Volume | | 500 | | | 500 | |
| Average Vol | | 125 | | | 125 | |

As an illustration of this concept, consider a hypothetical market with four stocks A, B, C and D, and assume that, on June 1st and October 1st, the trading volumes for the four stocks are as shown below:

The total volume traded on the two days is identical, but there are distinct differences in trading concentration. Trading on June 1st appears more evenly dispersed than on October 1st. For example, stocks with the two highest volumes on June 1st account for 56% of the total trading volume for the day, compared to the 78% for October 1st; at the low end, stocks with the two lowest volumes on June 1st account for 44% of the day's volume, compared to 22% on October 1st. To confirm the intuition that the distribution on June 1st is less concentrated than on October 1st, we employ the power law Eq (3) described above and estimate the TCI for both days. We find that the TCI for June 1st is a relatively low 0.28. However, for October 1st, the TCI is substantially higher at 1.35, confirming that trading on October 1st is more concentrated than on June 1st. This illustration shows how TCI provides a measure of the concentration of trading volumes. We use the same approach to compute and interpret the TCI for the distribution of trading volumes in S&P 500 index stocks and in other portfolios of stocks on the US markets.

## 3. Financial innovation and evolution of S&P 500 index funds

The S&P 500 index consists of five hundred large capitalization firms representing all sectors of the US economy. Together these 500 stocks represent more than 60% of the total market capitalization of all US equities covered in the Center for Research in Security Prices (CRSP) database. The index stocks are chosen by the Index Committee at Standard and Poor's according to established criteria that include industry/sector representation, domicile, financial viability, and market float. There is a substantial body of literature documenting the uniqueness of stocks included in or excluded from the S&P 500 index, starting with Shleifer [18]. This paper contributes to that literature by being the first to examine how the daily concentration of trading in the S&P 500 stocks has evolved over time.

The S&P 500 index was introduced in 1957 and was mostly used as a stock market indicator. It was not until 1975 that the first of the mutual funds designed to mimic the S&P 500 index was introduced. A S&P 500 index fund invests in the 500 individual stocks that comprise the S&P 500 index. The investments in each stock are proportional to their market capitalizations. Index funds offer investors the opportunity to invest in a large bundle or basket of screened stocks and gain the benefits of diversification with minimal effort. Since index fund

managers can operate passively, that is, they do not have to engage in an active search for attractive investments, they can also offer the additional benefit of lower expenses, a very important consideration for investors. This financial innovation of index funds, which was championed by Paul Samuelson and others, initially faced considerable skepticism from investment managers [10]. This new product innovation, however, has gained significant traction over the last 43 years and now accounts for over $3.4 trillion in indexed assets as reported in the August 2019 S&P 500 Factsheet [19]. This success may be attributed, at least in part, to the attractiveness of product bundling as studied in the economics and marketing literature [e.g., 20–22].

Prior to the introduction of index funds, investors would have to actively pick individual stocks listed in the S&P 500 index in an *a la carte* fashion. This method of actively selecting stocks would have been similar to the way that investors would have actively selected stocks that were not in the index. Therefore, we expect that the distribution of trading in S&P 500 stocks and the associated TCI (measuring trading concentration) would be similar to those for non-S&P 500 index stocks. However, the introduction of index funds and the entry of index fund investors would have changed the nature of trading in S&P 500 stocks. These index investors invest in the entire bundle of stocks that comprise the S&P 500 index in proportion to their market value, irrespective of the valuation of an individual stock in the index. In contrast, active investors would continue to actively pick individual stocks *a la carte* in the S&P 500 index based on the intrinsic value or criteria other than the stocks' membership in the index.

In an environment where all trading in S&P 500 stocks is done exclusively by index fund investors we would expect that the TCI would stay constant over time, since these investors would be trading the stocks in the same relative proportions every time. On the other hand, if trading in S&P 500 stocks is done exclusively by active stock pickers, we would expect the TCI to follow the same trends as the TCI for non-S&P 500 index stocks. If, as is typical, both types of investors are present, (i.e., *a la carte traders* as well as bundle traders) we expect to see the TCI change over time, with the change being driven by the dominant group of traders. There is, in addition, the possibility that the relative dominance of the two types of traders will vary over time, resulting in no distinct trend in the TCI for S&P 500 stocks. To discern among these competing hypotheses, we model the distribution of daily trading volume for both S&P 500 stocks and non-S&P 500 stocks as a power law function. We then study the trends in the TCI in the periods before and after the introduction of index funds to examine whether the introduction of index funds have affected the distribution of trading volumes in S&P 500 stocks. To flesh out our analysis, we examine whether the TCI of the non-S&P 500 stocks behave the same as the TCI for the entire market (i.e., all CRSP stocks).

We also provide a counterfactual analysis by examining the evolutionary behavior of the TCI of other sets of 500 stocks. First, we construct and analyze eight different portfolios each comprised of 500 stocks drawn randomly from CRSP. Second, we examine the behavior of the top 500 stocks i.e., those with the largest dollar trading volume. Third, we look at the S&P 500 index stocks with a different lens. Instead of using dollar trading volume (i.e., the product of shares traded and closing price) we now use market capitalization (i.e., the product of shares outstanding and closing price) for each of the firms in the index. This comparative analysis allows us to discern whether our intriguing findings of the TCI pattern of the inverted V-shape for S&P 500 stocks is unique or representative of the pattern for other sets of 500 stocks. If the S&P 500 TCI is the only one among these that has a unique pattern from the point of inflection at the introduction of index funds, we can then conclude this pattern derives from the influx of new index traders for this new product in the financial markets.

## 4. Analysis and results

Our analysis covers 14,850 trading days from January 1960 through December 2018. For each trading day in this period we select all the stocks that are in the Center for Research in Security Prices (CRSP) database. For each of the firms in the sample, we obtain the closing price and the total number of shares traded during the day and compute the dollar volume of trading in each stock for each day and rank them within a portfolio; we exclude stocks with zero volumes. With this data, we then compute the daily Trading Concentration Index, i.e., the power law exponent, using Eq (3) above for each of the portfolios discussed in the prior section.

We first compute the daily TCIs for the S&P 500 index portfolio. We contrast this to the daily TCI of the portfolio comprised of all other stocks (Non-S&P 500). We benchmark the TCI of the S&P 500 stocks and the non-S&P 500 stocks to the TCI for the entire market (i.e., all CRSP stocks). Next, to determine if the effects associated with S&P 500 membership merely reflect large firm size, we construct a portfolio of the 500 largest stocks by market capitalization, and compute the daily TCI for that portfolio (Top 500). If the trends that we observe with the S&P 500 portfolio are merely an artifact of firm size, we should see similar trends with the portfolio of the largest 500 stocks. As another comparison point, we construct eight portfolios of 500 stocks chosen randomly from the CRSP database (Random500 portfolio) and compute the average daily TCI for the distribution of the trading volumes in these eight portfolios.

### Results

A summary of the average dollar daily trading volumes for the S&P 500 and the non-S&P 500 portfolios is presented in Table 1. As shown there, dollar trading volumes in S&P 500 stocks grew at an annual rate of 14% from 1960 to 2018, from an average of $75 million per day in 1960 to an average of $168 billion per day in 2018. Dollar trading volume in non-S&P stocks grew at an even higher annual rate of 15.5% over this period, from an average volume of $38 million per day in 1960 to $186 billion per day in 2018. For the sake of brevity, we present the results for a selection of years from 1960 to 2018 in Table 1.

Table 1 also shows the trading volumes of the top quintile of stocks in the S&P 500 and in the non-S&P 500 categories, and their share of the total trading volume for all stocks in each category. We plot these quintile shares in Fig 1 and show that the share of the top quintile in non-S&P 500 stocks steadily increased from around 70% of all trading volume in the 1960's to more than 90% of all trading volume in recent years. This suggests that, over time, trading in non-S&P 500 stocks has become more and more concentrated in the top stocks. However, we find a different trend in the case of the S&P 500 stocks. We find that the percentage share of the top quintile in the total trading volume for S&P 500 stocks has declined from 70% in the mid-1970's, and has been in the 60% range in the last few years, suggesting a decrease in the concentration of trading in the top quintile of the S&P 500 stocks.

To more formally examine the apparent change in the concentration of trading, we model the entire distribution of daily trading volumes for the S&P 500 and the non-S&P 500 stocks as a power law function and estimate the daily TCI using Eq (3) described earlier. We estimate the daily TCI for each of the 14,849 trading days in the 1960 to 2018 period. In Fig 2, rather than depict all 14,849 data points, we only plot the annual averages of the daily TCI from 1960 to 2018. This provides a cleaner and clearer picture of how the concentration of trading in these categories has changed over the last six decades.

Table 1. Average daily trading volume of S&P 500 stocks and non-S&P 500 stocks and the trading volume of the top quintile of stocks in each group, for select years from 1960 to 2018.

| Year | S&P 500 Stocks | | | Non-S&P 500 Stocks | | | |
|------|-------------------------------------------------|----------------------------------------------|-----------------------------------------------|--------------------------------|-------------------------------------------------|----------------------------------------------|-----------------------------------------------|
| | Average Daily Trading Volume (in $ millions) | Average Volume of the Top Quintile Stocks | Top Quintile Volume as % of Total Volume | Average Number of shares traded | Average Daily Trading Volume (in $ millions) | Average Volume of the Top Quintile Stocks | Top Quintile Volume as % of Total Volume |
| 1960 | $75 | $51 | 68% | 553 | $38 | $28 | 75% |
| 1964 | $136 | $95 | 70% | 1391 | $74 | $59 | 80% |
| 1968 | $326 | $218 | 67% | 1655 | $364 | $271 | 74% |
| 1972 | $342 | $237 | 69% | 2015 | $310 | $251 | 81% |
| 1976 | $529 | $369 | 70% | 1880 | $226 | $192 | 85% |
| 1980 | $1,166 | $799 | 69% | 1773 | $613 | $513 | 84% |
| 1984 | $2,857 | $1,945 | 68% | 4940 | $1,283 | $1,133 | 88% |
| 1988 | $5,251 | $3,541 | 67% | 5329 | $2,295 | $2,071 | 90% |
| 1992 | $6,664 | $4,376 | 66% | 5387 | $4,926 | $4,391 | 89% |
| 1996 | $16,725 | $11,403 | 68% | 7524 | $14,985 | $13,357 | 89% |
| 2000 | $69,858 | $54,586 | 78% | 7326 | $66,098 | $62,882 | 95% |
| 2004 | $53,550 | $33,424 | 62% | 5981 | $47,562 | $42,531 | 89% |
| 2008 | $131,354 | $83,493 | 64% | 6221 | $165,660 | $157,137 | 95% |
| 2012 | $102,577 | $63,700 | 62% | 5995 | $106,455 | $98,762 | 93% |
| 2014 | $116,043 | $68,551 | 59% | 6297 | $140,025 | $127,293 | 91% |
| 2016 | $125,586 | $70,814 | 56% | 6447 | $142,677 | $130,697 | 92% |
| 2018 | $168,376 | $104,243 | 62% | 6695 | $185,793 | $170,502 | 92% |

As shown in Fig 2, for the period 1960 to 2018, the TCI has steadily increased for non-S&P 500 stocks from around 1.5 to slightly above 3.0. However, the TCI for the S&P 500 stocks exhibits a remarkably different trend. After tracking the non-S&P 500 stocks in the early years, the TCI for the S&P 500 stocks markedly diverge after the mid-1970's, and steadily decline to levels around 1.0 in recent years. This inflection and resultant divergence in the TCI suggests that while concentration of trading in the non-S&P 500 stocks has continually increased over time, the concentration of trading in S&P 500 stocks has steadily declined after reaching a peak in the mid-1970's at about the time when the first S&P 500 index funds started to gain popularity.

To confirm the divergence between the two series of concentration indices, we plot their difference in Fig 3. As this chart shows, the difference between the trading concentrations for the S&P 500 and the non-S&P 500 stocks remained small or zero until 1975 but started an upward climb thereafter. We confirm this visual observation by estimating the slope of the difference line for the two time periods; 1960 to 1975, and 1976 to 2018. For the 1960–1975 period, the slope ($0.08 \times 10^{-5}$) is statistically insignificant ($t = 0.4$; $R^2 = 0$) and is indistinguishable from zero. However, for the 1976 to 2018 period, the slope ($19.9 \times 10^{-5}$) is a statistically significant ($t = 371$; $R^2 = 0.93$).

## Counterfactual analysis

It is possible that the declining concentration in S&P 500 stocks simply reflects trading patterns unique to large capitalization firms. To rule out this possibility, we construct a portfolio of the top 500 stocks by daily market value from the CRSP database and compute its daily TCI. In Fig 4, we graph the TCI of the S&P 500 stocks and the TCI of the Top 500 stocks. We find that the TCI for the Top 500 stocks have remained fairly stable, hovering around a value of 1.0 over the last 55 years. Results for the Top-500 stocks are eerily reminiscent of the power-law

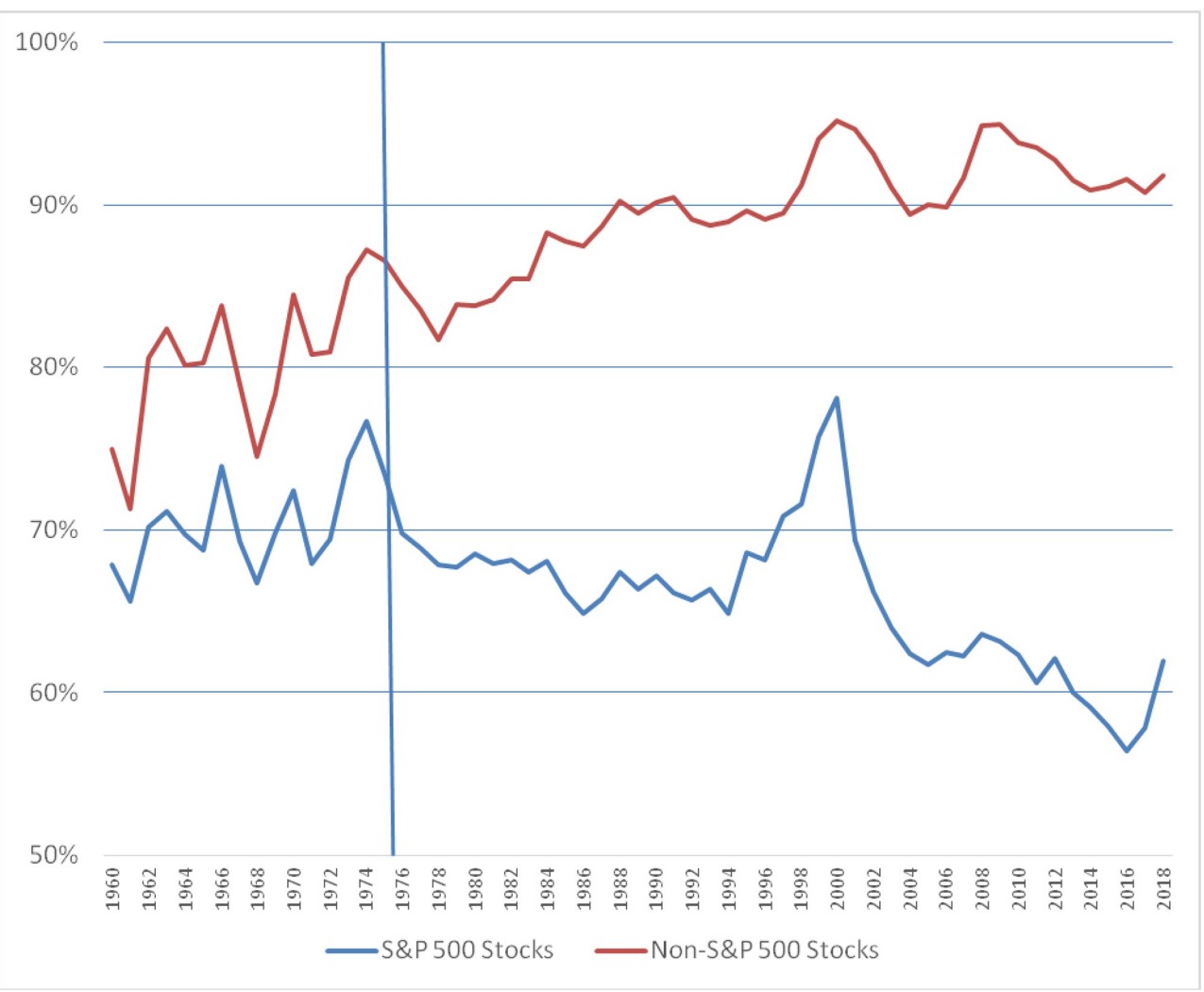

**Fig 1. Percentage share of the top quintile of stocks in the total trading volume of S&P 500 and non-S&P 500 stocks, 1960 to 2018.**

exponent values of 1.0 for the top ranked (by population) cities across time and countries. This classic example of Zipf's law is noted as one of the strongest and most puzzling empirical generalizations in the economics literature [14,23].

This flat line for the TCI of the Top-500 is in sharp contrast to the S&P 500's inverted V-shaped trend. It would therefore appear that the decrease in the concentration of trading in S&P 500 stocks after 1975 appears to be driven by factors other than purely market capitalization. As an additional test of the uniqueness of the S&P 500's trend, we construct eight portfolios of 500 stocks drawn at random each day from the overall market and compute the daily TCI for each of these portfolios. The average daily TCI for these eight draws of random 500 stock portfolios (Random500) are also shown in Fig 4. Here again, we observe that the trend for the Random500 stock portfolios is different from the trend for S&P 500 stocks; the trend for the Random500 follows the non-S&P 500 stocks and increases steadily over time. Further, the Random500 tracks the trend of the portfolio of the entire bundle of CRSP stocks (AllStks) almost precisely.

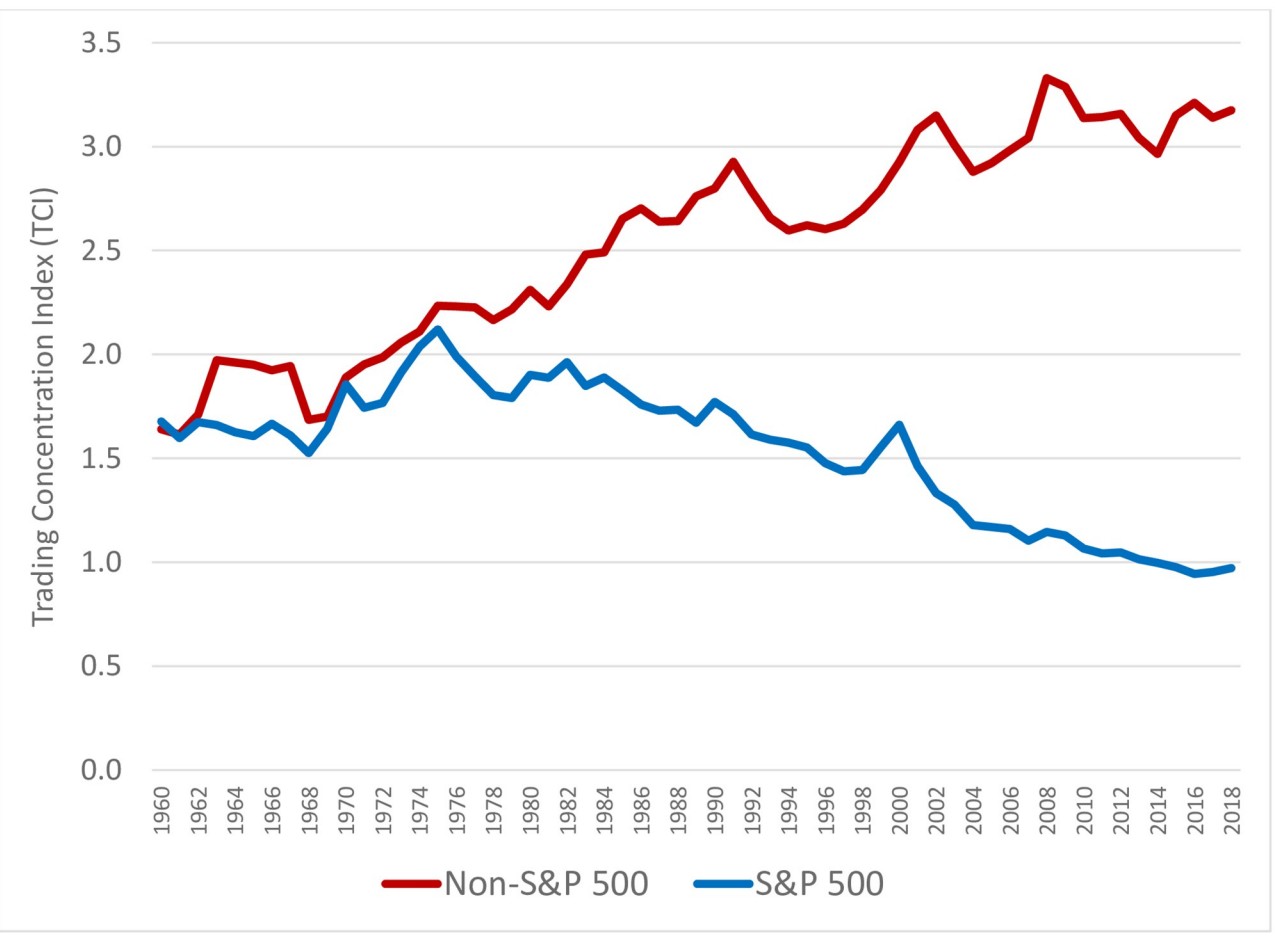

**Fig 2. Annual averages of the daily Trading Concentration Indices (power low exponents) for S&P 500 stock portfolios and non-S&P 500 stock portfolios, 1960 to 2018.**

The visual trends in Fig 4 clearly suggest that trading patterns in the S&P 500 stocks are unique as it is the only portfolio examined that displays a downward trend after 1975. To confirm the statistical significance of these visual trends depicted in Fig 4, we regress the TCI against calendar time for each of the portfolios described above. For these regressions, we consider two time periods: the period before the introduction of index funds (1960–1975), and the period after index funds were introduced (1976–2018). The summary results of these regressions are in Table 2.

As shown in Table 2 and consistent with the visual trend lines, the slope estimates for the Non-S&P 500 stocks are significantly positive and similar in both the 1960–75 and the 1976–2018 time periods ($10.62 \times 10^{-5}$ and $9.43 \times 10^{-5}$, respectively). However, we see a distinct change between the two periods for the S&P 500 stocks. In the 1960–1975 period, the slope estimate for S&P 500 stocks is significantly positive ($10.54 \times 10^{-5}$) and almost identical to the Non-S&P 500 stocks slope estimate. However, in the 1976–2018 period, the S&P 500 stocks slope estimate ($-10.49 \times 10^{-5}$) turns significantly negative and is substantially different from the positive slope for the Non-S&P 500 stocks. For the Top500 stocks, there is no discernible trend over time, as is evident from the near-zero slope coefficients ($1.84 \times 10^{-5}$; $-1.04 \times 10^{-5}$, respectively) for the two time periods. This confirms the visual evidence from Fig 4 that the TCI for the Top500 stocks remains relatively flat for the entire 1960 to 2018 period. Finally, we

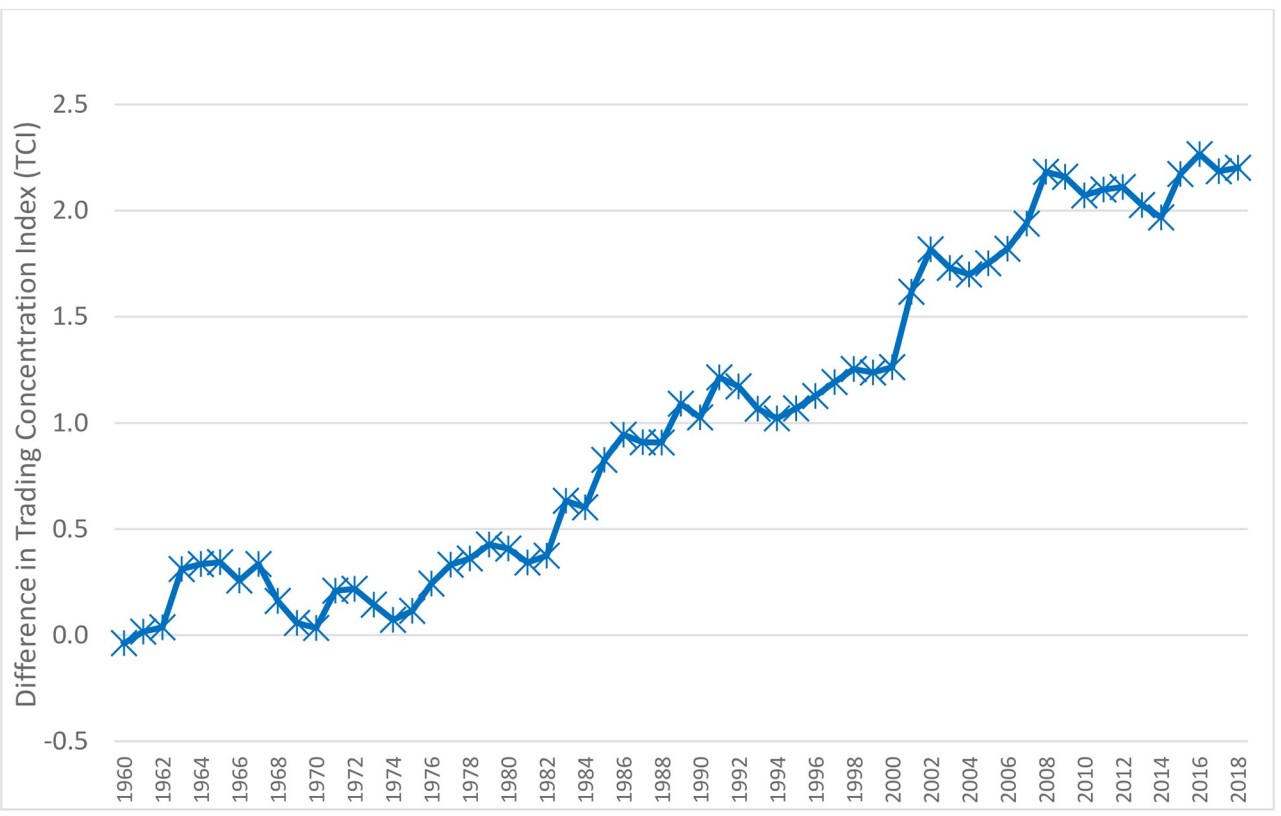

**Fig 3. Different between the annual averages of the daily Trading Concentration Indices for S&P 500 and non-S&P 500 portfolios, 1960 to 2018.**

observe that the slope estimates for the Random500 stocks mirror the non-S&P 500 stocks in both time periods in showing a steady increase in trading concentration.

As an additional robustness check for changes in the trading patterns after 1975, we conduct a segmented or piecewise regression using a dummy variable that takes the value '0' for the 1960–75 period and '1' for the 1976–2018 period. We then compare the regression fit measures of this segmented regression model with the corresponding measures of a simple linear model over the whole period (1960 to 2018). If there is a change after 1975, we would expect the segmented regression to offer better regression fit measures—higher coefficient of correlation and lower standard errors–than the simple linear model.

We present the regression measures for the segmented regression model and the simple linear model in Table 3. As shown there, the coefficient of correlation (R-squared) for the segmented regression for the S&P 500 portfolio is substantially higher at 88% when compared to the 60% in the simple linear model; the standard error of the segmented model is substantially lower at 0.12 compared to 0.21 in the simple linear model; and, the F-Stat for the segmented regression is much higher than the F-Stat for the simple linear model. All these measures suggest that the segmented regression model, with a break occurring after 1975, provides a much better fitthan the simple linear model for the S&P 500 portfolio. In contrast, in the case of Non-S&P 500 portfolio, the Top 500 portfolio and the Random 500 portfolio, there is very little difference between the R-Squared and standard errors of the segmented regression and the simple linear regression; this suggests that, with these portfolios, segmenting the model after 1975 does not affect the model estimates. Overall, this comparison appears to provide strong

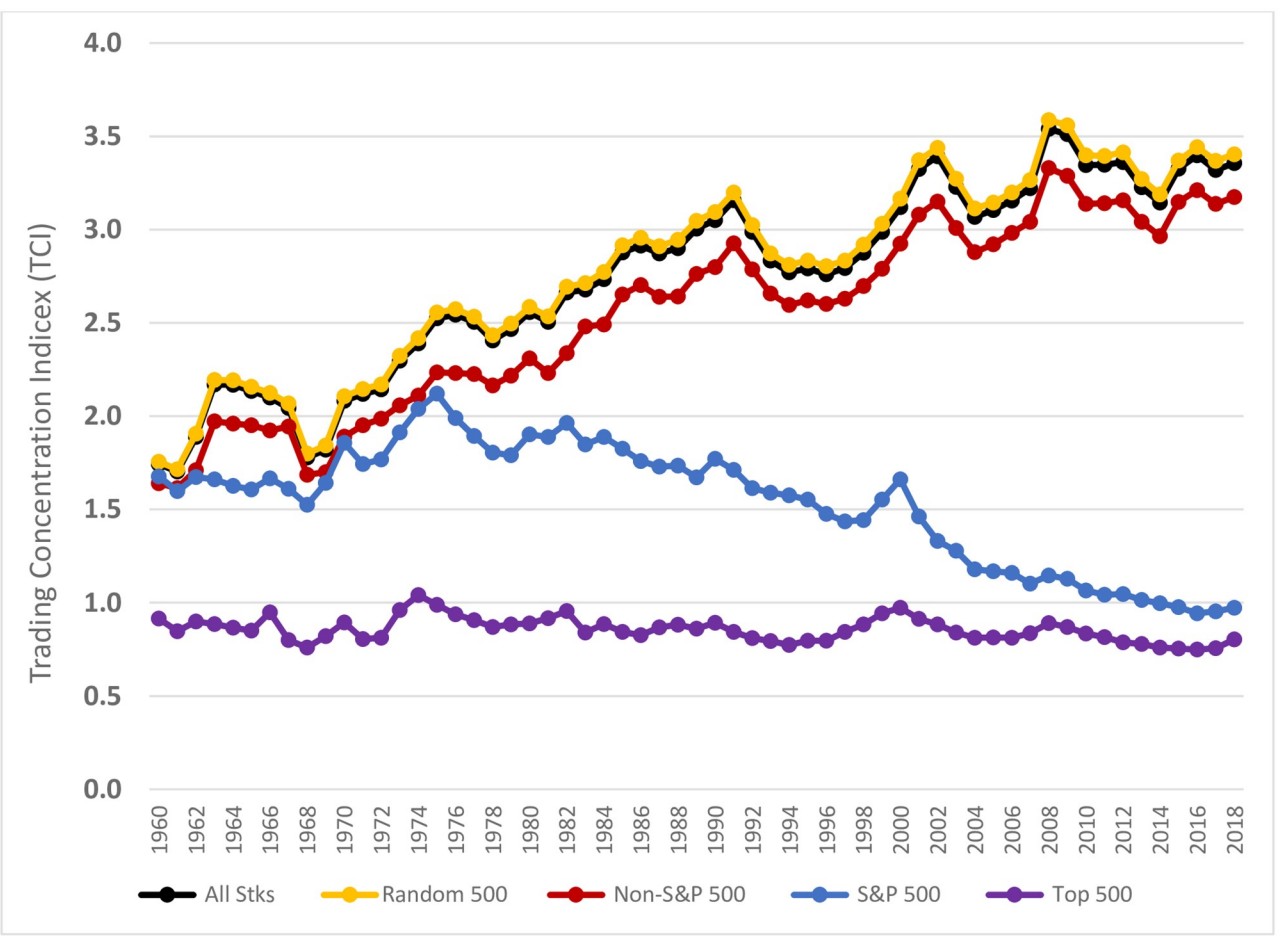

**Fig 4. Annual averages of daily Trading Concentration Indices (TCI) for all stocks, top 500 stocks, random500 stocks, S&P 500 stocks, and non-S&P 500 stocks portfolios, 1960 to 2018.**

supporting evidence that the trading concentration for S&P 500 stocks declined after 1975, but there is no similar decline for the other portfolios examined.

## Index investors vs. active investors

The results in Table 2 suggest that the entry of index fund investors in 1976 set in motion the gradual change towards more uniform trading in S&P 500 index stocks. Over time, their trading of the S&P 500 as a bundle has gained dominance over traditional investors who pick individual S&P 500 stocks in an *al a carte* fashion. We see no such evidence of similar changes in

**Table 2. Regression slope estimates of Trading Concentration Indices against calendar time for different stock portfolios, 1960 to 2018.**

| Portfolios | 1960 to 1975 (N = 4,005 Trading Days) | | | 1976 to 2018 (N = 10,844 Trading Days) | | |
|---|---|---|---|---|---|---|
| | Slope estimate (x $10^{-5}$) | t-value | $R^2$ | Slope estimate (x $10^{-5}$) | t-value | $R^2$ |
| Non-S&P 500 stocks | 10.62 | 52 | 0.40 | 9.43 | 193 | 0.77 |
| S&P 500 stocks | 10.54 | 57 | 0.50 | 10.49) | (320) | 0.90 |
| Top 500 stocks | 1.84 | 15 | 0.05 | (1.04) | (57) | 0.23 |
| Random500 stocks | 14.08 | 54 | 0.40 | 8.79 | 164 | 0.71 |

**Table 3. Comparison of regression fit measures for segmented regression and simple linear regression models, with Trading Concentration Indices modelled against calendar time from 1960 to 2018.** In the segmented regression, the dummy variable has a value of '0' from 1960 to 1975 and '1' from 1976 to 2018.

| Portfolio | Regression model | R-Square | Std Error of Model | F-Stat for Regression |
|---|---|---|---|---|
| S&P 500 | Segmented Reg | 0.88 | 0.12 | 52,393 |
| | Simple Linear reg | 0.60 | 0.21 | 22,491 |
| Non-S&P500 | Segmented Reg | 0.90 | 0.16 | 63,503 |
| | Simple Linear reg | 0.89 | 0.17 | 119,384 |
| TOP500 | Segmented Reg | 0.19 | 0.07 | 1,779 |
| | Simple Linear reg | 0.13 | 0.07 | 2,258 |
| Random500 | Segmented Reg | 0.88 | 0.18 | 52,116 |
| | Simple Linear reg | 0.86 | 0.20 | 87,917 |

the trend or composition of the type of traders for the Non-S&P 500 stocks, or the other five hundred stock portfolios, i.e., the Top500 or the Random500.

The regression results confirm a gradual change towards more even uniform trading in S&P 500 index stocks strongly suggesting this change was due to the emergence of index fund investors after 1975. Over time, it would appear that index traders have gained dominance over the active investors in trading S&P 500 stocks. To further examine this idea, we estimate the TCI based on the market capitalizations of the stocks in the S&P 500 and compare this with the TCIs for the dollar trading volumes of the S&P 500 stocks. Our hypothesis is that if all the trading in S&P 500 stocks were done only by index investors, the TCI based on the market capitalizations should be identical to TCI based on trading volumes. We should expect to see this because index investors trade the S&P 500 stocks as a bundle in proportion to their market capitalization. Alternatively, if active investors (i.e., *a la carte* stock pickers) played a larger role in daily trading, we would expect the TCI for the trading volume distributions to be higher than the TCI for the market capitalization distributions. We should expect this alternative hypothesis because active investors, by focusing on only a few stocks within the S&P 500 index, would contribute to more uneven distribution of trading volume.

To depict this more clearly, in Fig 5 we show the TCI trends for the S&P 500 based on both the dollar trading volume as well as the market capitalization metric during 1976–2018 period. i.e., the index funds era. In the early years of this era, the dollar trading volume TCI are substantially higher than those for the market-capitalization measure. This indicates that the dollar trading volumes were significantly more concentrated in a few of the more popular S&P stocks. This is possibly due to the dominant role of active investors engaging in *a la carte* trading of S&P 500 stocks. In fact, using the market capitalization of the S&P 500 stocks as the metric to rank daily trading volume, we even find a surprising gradual downward trend in trading concentration from 1975–2018. Yet, over time, the gap between the two series of TCI has dramatically decreased and, in fact, the two series have in recent times more or less converged. Interestingly, the convergence is at a power law exponent of around 1.0 indicating another manifestation of Zipf's Law. This convergence suggests that the volume of daily trading in S&P 500 stocks has become almost exactly proportional to the market cap of the stocks that make up the index. This leads us to conclude that trading in S&P 500 stocks is now almost completely dominated by index traders.

Our results using market capitalization are contrary to recently expressed views in the popular press regarding trading concentration which assert that trading is now more concentrated in only a few stocks [9]. To be specific, we found that in 1960, the total market cap of the top 100 stocks within the S&P 500 was about 84 times the total market cap of the bottom 100 stocks within the S&P 500; in 2018, this number had dropped to about 18 suggesting more even trading across all 500 stocks in the index.

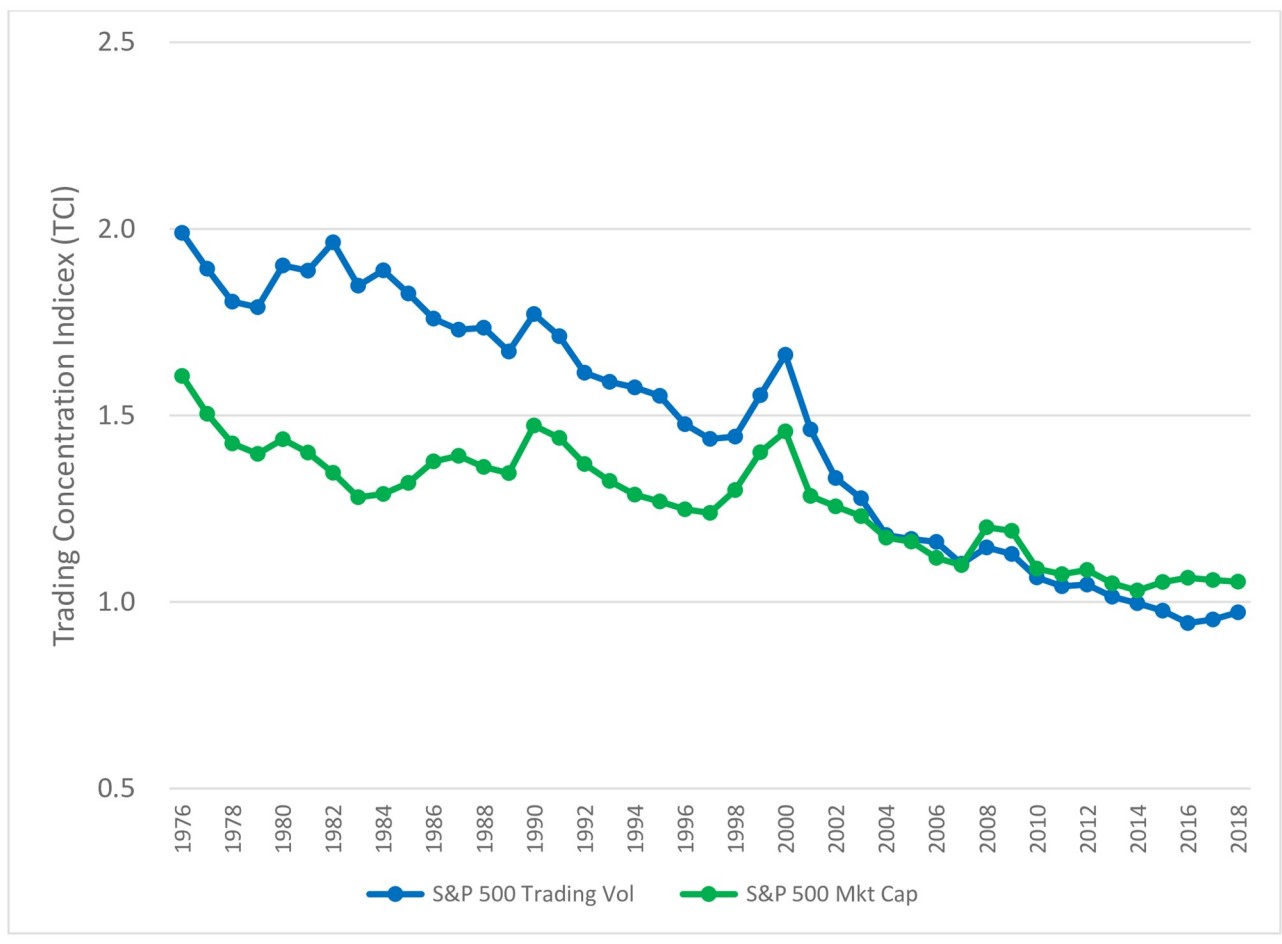

**Fig 5. Annual averages of the daily Trading Concentration Indices for the distribution of S&P 500 trading volumes and S&P 500 market capitalizations.**

## 5. Conclusion and future directions

In this paper, we study the changes in the distribution of daily trading volumes of stocks in the CRSP database over the last six decades. We model the distribution of daily trading volumes as a power law function and use the power law exponent or TCI as a measure of trading concentration. For the vast number of stocks that are not part of the S&P 500 index, as well as for the entirety of CRSP stocks, we find that the Trading Concentration Indices (TCI) have steadily increased from 1960 to 2018. This suggests that daily trading is becoming more and more concentrated in a small subset of stocks. The phenomenon of an increasing trend in the concentration of daily trading volumes (in number of shares traded) for the period 1962–2005 was first rigorously documented in Balakrishnan, et al [1]. This result held for the entire CRSP database as well as for stocks on each of the three major US stock exchanges. Their finding, consistent with Zipf's Principle of Least Effort, suggests that investors exhibit "herding behavior", i.e., overwhelmed by a plethora of stock investment offerings they limit their daily choices to only a small cohort of stocks. We should point out that our revalidation of the phenomenon in this paper both encompasses a significantly longer time period (1960–2018) and employs a different but similar metric (i.e., daily dollar trading volume). This belief in the increasing concentration in a short list of stocks now seems to have become fairly well accepted by equity and quantitative analysts as reported in the popular press [9].

In the mid-1970's a new product innovation in financial products was first offered to investors, namely S&P 500 index funds. These funds were purposefully designed to allow investors to trade the entire bundle of S&P 500 stocks. It has been an open question as to whether this bundle trading impacted "herding behavior" (which results in focusing on only a small subset of stocks) in any way. Here, our examination of this question leads to an extremely intriguing empirical result. Contrary to the current belief of equity analysts, we find that since the introduction of this financial innovation, trading concentration has become more even across all stocks in the S&P 500 Index. This paper thus extends the literature on product bundling to the domain of stock markets and financial innovations.

More specifically, we find that the concentration of trading in S&P 500 stocks is distinctly different from the rest of the US market. The TCI for S&P 500 stocks rise along with the non-S&P 500 stocks from 1960 to 1975; however, since the introduction of the bundled product in 1975 the TCI of S&P 500 stocks have diverged and steadily declined. The uniqueness of this inverted v-shaped pattern is further corroborated by our counterfactual investigation of the TCI patterns for other sets of 500 stock portfolios. Our analysis shows that since 1975 trading in S&P 500 stocks has increasingly become more uniform across the 500 stocks that make up the index. This downward trend in the TCI is consistent with the hypothesis that S&P 500 index fund investors, who trade all stocks in the index as a bundle, have gradually become the dominant traders in S&P 500 stocks over the last 40 odd years. In the process, it appears that active investors have significantly less of an impact on the distribution of daily trading volume in S&P 500 stocks.

An additional finding that was unexpected and unhypothesized was the gradual downward trend in trading concentration from 1975–2018 when using the market capitalization of the S&P 500 stocks as the metric to rank daily trading volume. The reasons for this downward trend that we document in this paper are questions for future research. Specifically, it is not clear whether the downward trend implies that all stocks are growing in trading volume, but that smaller stocks are growing at a greater rate. Or, whether growth in trading volume for small stocks results from a substitution effect with investors shifting trading away from the larger stocks. But if this trend is due to the substitution effect, it is not happening at the level of the entire market (i.e., all CRSP) where daily dollar trading volume has become more uneven over the last many decades.

It would also be interesting to conduct a similar analysis on the evolution of trading concentration of other indices. For instance, whether the introduction of bundled products linked to indices such as the Russell 1000, S&P MidCap 400 Index, and the S&P SmallCap 600 Index resulted in similar changes in trading concentration as documented here for the introduction of S&P 500 Index funds. Potentially, a similar investigation for international stock indices and their associated bundled products may lead to empirical generalizations or other intriguing conclusions. It would also be very interesting to examine the impact of trading concentration evolution in S&P 500 stocks on the price discovery mechanism of index stocks, and its effect on the "security-level analysis that is required for true price discovery" [24]. Another important avenue for future research is building an analytical model, along the veins of those developed for the population distribution of the largest cities, to help understand what processes might lead the Top500 stocks to follow Zipf's law.

## Author Contributions

**Conceptualization:** S. Gowri Shankar, James M. Miller, P. V. (Sundar) Balakrishnan.

**Data curation:** S. Gowri Shankar.

**Formal analysis:** S. Gowri Shankar.

**Investigation:** S. Gowri Shankar.

**Methodology:** S. Gowri Shankar, P. V. (Sundar) Balakrishnan.

**Project administration:** S. Gowri Shankar, James M. Miller, P. V. (Sundar) Balakrishnan.

**Resources:** S. Gowri Shankar.

**Software:** S. Gowri Shankar.

**Supervision:** S. Gowri Shankar, James M. Miller, P. V. (Sundar) Balakrishnan.

**Visualization:** P. V. (Sundar) Balakrishnan.

**Writing – original draft:** S. Gowri Shankar, James M. Miller.

**Writing – review & editing:** S. Gowri Shankar, James M. Miller, P. V. (Sundar) Balakrishnan.

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
