## [Decision Letter · Decision Letter 0]

17 Jan 2020

PONE-D-19-27734

Evolutionary Disruption of S&P 500 Trading Concentration: An Intriguing Tale of a Financial Innovation

PLOS ONE

Dear Prof. Miller,

Thank you for submitting your manuscript to PLOS ONE. After careful consideration, we feel that it has merit but does not fully meet PLOS ONE’s publication criteria as it currently stands. Therefore, we invite you to submit a revised version of the manuscript that addresses the points raised during the review process.

We would appreciate receiving your revised manuscript by Mar 02 2020 11:59PM. To enhance the reproducibility of your results, we recommend that if applicable you deposit your laboratory protocols in protocols.io, where a protocol can be assigned its own identifier (DOI) such that it can be cited independently in the future. For instructions see: http://journals.plos.org/plosone/s/submission-guidelines#loc-laboratory-protocols

We look forward to receiving your revised manuscript.

Kind regards,

Jichang Zhao, Ph.D.

Academic Editor

PLOS ONE

Reviewers' comments:

Reviewer's Responses to Questions

**Comments to the Author**

1. Is the manuscript technically sound, and do the data support the conclusions?

Reviewer #1: Yes

Reviewer #2: Yes

2. Has the statistical analysis been performed appropriately and rigorously? 

Reviewer #1: Yes

Reviewer #2: Yes

3. Have the authors made all data underlying the findings in their manuscript fully available?

Reviewer #1: Yes

Reviewer #2: Yes

4. Is the manuscript presented in an intelligible fashion and written in standard English?

Reviewer #1: Yes

Reviewer #2: No

5. Review Comments to the Author

Reviewer #1: This paper examines the distribution of daily trading concentrations and discovers unique patterns before and after the introduction of S&P index funds. The paper is written well in many ways and the findings are interesting and insightful.

I have a few comments as follows.

1. Equation (1) and (2) are not in the right formats, as well as formula on page 10 and others.

2. Why is equation (1) a generalization of the power law distribution of volume? The author has well explained why the power law exponent could be used as a measure of market concentration, but not equation (1), which is used in the rest of this paper. I suggest authors give further explanation in the paper.

3. I also suggest the authors give a summary on how index fund trade in section 3, instead of fragmented descriptions throughout the discussion, to give a overview on how the index funds trading behavior can influence the trading concentration on S&P 500 stocks.

4. In the counterfactual analysis, the regression slope for the S&P 500 stocks is very close to zero, though it is of statistical significant. Therefore, I have doubts on the conclusion that the findings is not a unique pattern for large capitalization firms but for the trading behavior of index funds.

5. Also, other regression analysis like regression with dummy variable of time period is expected to be included in the counterfactual analysis, as an alternative of conducting two regression models for two time periods.

6. In section “index investors vs active investors”, I understand that“This convergence suggests that the volume of daily trading in S&P500 stocks has become almost exactly proportional to the market cap of the stocks that make up the index. ”, but why this statement leads to the conclusion that “This leads us to conclude that trading in S&P 500 stocks is now almost completely dominated by index traders” ? Even though the index investors may trade the stocks as a bundle in proportion to their market capitalization, this cannot exclude the possibility that other investors trade this way, especially when we consider herding behaviors.

Reviewer #2: Review of Manuscript No: PONE-D-19-27734

Manuscript Title: Evolutionary Disruption of S&P 500 Trading Concentration: An Intriguing Tale of a Financial Innovation.

Summary:

This paper examines the trading concentration in financial markets surrounding the introduction of S&P 500 index funds. In particular, the paper models the distribution of daily trading concentration as a power law function and examines the function’s exponents as a measures of trading concentration. While trading concentration has increased for the entire market over the last six decades, trading concentration in S&P 500 stocks has steadily decreased after index funds (tracking the S&P 500) have become available. These results appear robust to randomly selected portfolios of non-S&P 500 stocks. The results are supportive of the idea that index funds have allowed S&P 500 stocks to be more regularly traded due to the innovation of index funds.

Comments:

The paper offers a potentially interesting set of findings. The empirical analysis follows a set a studies that have examined trading concentration and seems to be both appropriate and correct. However, I have a few comments regarding the exposition of the paper that might broaden the overall contribution of the study.

1. Periodically, the paper describes trading unevenness and trading concentration. The paper needs to provide careful context when describing these measures of trading activity. Consistency in the exposition could help the readability of the paper. For instance, upfront the paper could define trading concentration and refer to this term throughout the study as opposed to using alternative descriptions.

2. The paper needs to more carefully describe how the empirical methods capture trading concentration. The description of the methods is appropriate, but providing a better link between the methods and the measure of trading concentration might be needed.

3. There are a couple of statements that could be debated. First, on page 1, the introduction suggests that there is a rapidly expanding set of stock offerings. Others (Gao, Ritter, and Zhu (2013); Doidge, Karolyi, and Stulz (2013, 2017)) argue that stock offerings are declining. This statement, therefore, needs to be better justified. Second, the introduction assumes that investors are overwhelmed by the large number of investment choices. This may be true but this statement also needs further justification. Third, the paper infers that the innovation of index funds was in response to resolve the problem of too many investment choices. This statement seems to be a little strong.

4. There are several grammatical issues throughout the study. Below I have provided a non-exhaustive list:

a. On page 2, it should read, “After the introduction, we expect …”

b. On page 2, it should read, “If trading volume in S&P 500 index funds increases over time…”

c. On page 11, it should read, “…by estimating the power law exponent using equation (3)…”

d. On page 11, it should read, “Next, to determine if the effects associated…”

Again, this is a non-exhaustive list. I would recommend that the authors provide a series of careful edits to correct any other grammatical issues.

6. PLOS authors have the option to publish the peer review history of their article (what does this mean?). If published, this will include your full peer review and any attached files.

Reviewer #1: No

Reviewer #2: No

---

## [Author Response · Author response to Decision Letter 0]

26 Feb 2020

PONE-D-19-27734 - Revision Notes

PONE-D-19-27734

Evolutionary Disruption of S&P 500 Trading Concentration: An Intriguing Tale of a Financial Innovation

PLOS ONE 

Revision Notes: Response to Reviewers

Comments for EDITOR 

We greatly appreciate your positive response to our manuscript and your recommendation to revise the manuscript. The reviewers’ clear comments provided us with a roadmap for undertaking our revision. Our sincere thanks. We have attached detailed replies to the comments made by both Reviewers. Responses to each of the issues raised are provided below. Thank you again for the opportunity to revise the paper. 

Responses for Reviewer #1 

Reviewer #1: This paper examines the distribution of daily trading concentrations and discovers unique patterns before and after the introduction of S&P index funds. The paper is written well in many ways and the findings are interesting and insightful.

Response: We thank you for your positive comments. We are very gratified that you found the findings interesting and insightful. We truly appreciate your close reading of the manuscript. We have revised the paper based on your detailed feedback below. We have provided a set of notes below that describes our work to address your comments and as to how it is has been incorporated into the paper. We believe that as a result it is a much-improved paper. Thank you.

Reviewer: I have a few comments as follows.

1. Equation (1) and (2) are not in the right formats, as well as formula on page 10 and others.

Response: We have now revised them. Thank you.

2. Why is equation (1) a generalization of the power law distribution of volume? The author has well explained why the power law exponent could be used as a measure of market concentration, but not equation (1), which is used in the rest of this paper. I suggest authors give further explanation in the paper.

Response: Thank you. We have now revised the earlier description to make things clearer. We use the formulation for market concentration as specified by Balakrishnan, Miller and Shankar (Economics Letters, 2008). Based off this generalized formulation of the power law, we followed with the operationalization to the trading volume concentration. We believe these additions and explanations of the special case as it applies to trading volume address the substance and the spirit of the comment. 

3. I also suggest the authors give a summary on how index fund trade in section 3, instead of fragmented descriptions throughout the discussion, to give a overview on how the index funds trading behavior can influence the trading concentration on S&P 500 stocks.

Response: We thank you for this comment. In our revision, we have added a couple of paragraphs that summarizes the nature and trades in index funds. We agree with you and believe that this will be useful for the lay readers. However, in our rewrite, as we ran it by other friendly readers, we discovered that if we delete some of the description of index funds in other sections, then the paragraphs tend to lose their meaning and makes it harder for the reader to grasp the logic. We trust this addresses the spirit of your comment. 

4. In the counterfactual analysis, the regression slope for the S&P 500 stocks is very close to zero, though it is of statistical significant. Therefore, I have doubts on the conclusion that the findings is not a unique pattern for large capitalization firms but for the trading behavior of index funds.

5. Also, other regression analysis like regression with dummy variable of time period is expected to be included in the counterfactual analysis, as an alternative of conducting two regression models for two time periods.

Response: The preceding comments 4 & 5 are highly though provoking. Based on this we decided to conduct an additional set of statistical analyses. Additional analysis included running segmented regression models, i.e., piecewise linear regression, to assess the validity of the results as reported in Table 2. We now report some of those results in the paper (Table 3). Based on these new results, we are now more comfortable in our finding stating the uniqueness of the inverted V-shaped pattern for the S&P 500 Index stocks. We believe that reporting this additional analysis has made the paper stronger and the findings more robust. We truly thank you for this excellent suggestion.

6. In section “index investors vs active investors”, I understand that “This convergence suggests that the volume of daily trading in S&P500 stocks has become almost exactly proportional to the market cap of the stocks that make up the index. ”, but why this statement leads to the conclusion that “This leads us to conclude that trading in S&P 500 stocks is now almost completely dominated by index traders” ? Even though the index investors may trade the stocks as a bundle in proportion to their market capitalization, this cannot exclude the possibility that other investors trade this way, especially when we consider herding behaviors.

Response: We think this is indeed a valid observation. We have softened our language on this. Thank you.

 

Responses for Reviewer #2 

Summary:

This paper examines the trading concentration in financial markets surrounding the introduction of S&P 500 index funds. In particular, the paper models the distribution of daily trading concentration as a power law function and examines the function’s exponents as a measures of trading concentration. While trading concentration has increased for the entire market over the last six decades, trading concentration in S&P 500 stocks has steadily decreased after index funds (tracking the S&P 500) have become available. These results appear robust to randomly selected portfolios of non-S&P 500 stocks. The results are supportive of the idea that index funds have allowed S&P 500 stocks to be more regularly traded due to the innovation of index funds.

Response: We thank you for your very positive and encouraging comments. We are gratified that you found the findings interesting and insightful. We truly appreciate your close reading of the manuscript. We have revised the paper based on your detailed feedback below. We have provided a set of notes below that describes our work to address your comments and as to how it is has been incorporated into the paper. We believe that as a result it is a much-improved paper. Thank you.

Reviewer Comments:

The paper offers a potentially interesting set of findings. The empirical analysis follows a set a studies that have examined trading concentration and seems to be both appropriate and correct. However, I have a few comments regarding the exposition of the paper that might broaden the overall contribution of the study.

1. Periodically, the paper describes trading unevenness and trading concentration. The paper needs to provide careful context when describing these measures of trading activity. Consistency in the exposition could help the readability of the paper. For instance, upfront the paper could define trading concentration and refer to this term throughout the study as opposed to using alternative descriptions.

Response: This is an excellent point. We have now tried to do this as much as possible. This set of edits may help resolve some of the trouble with access to and consistency in the exposition. Thank you.

2. The paper needs to more carefully describe how the empirical methods capture trading concentration. The description of the methods is appropriate, but providing a better link between the methods and the measure of trading concentration might be needed.

Response: Our original concern was that we may have even overdone this explanation. We have now, therefore, at your suggestion, further strengthened our link to the original research of Naldi (2003), and Balakrishnan et al. (2008). We trust that this along with the example and the additional exposition detailed above resolves any remaining confusion and makes it easy for readers to follow.

3. There are a couple of statements that could be debated. First, on page 1, the introduction suggests that there is a rapidly expanding set of stock offerings. Others (Gao, Ritter, and Zhu (2013); Doidge, Karolyi, and Stulz (2013, 2017)) argue that stock offerings are declining. This statement, therefore, needs to be better justified. Second, the introduction assumes that investors are overwhelmed by the large number of investment choices. This may be true but this statement also needs further justification. Third, the paper infers that the innovation of index funds was in response to resolve the problem of too many investment choices. This statement seems to be a little strong.

Response: Thank you for this excellent point. We have clarified that the statement in our introduction about investors being overwhelmed by the large number of investment choices comes directly from an earlier 2008 paper that was examining the 1962 to 2005 period. However, as reported in recent literature, the number of stocks traded on stock exchanges have come down recently (though it is remains higher than the numbers in the 1960’s, the earliest point in our analysis). A detailed footnote to address this issue has been added to the paper, along with appropriate edits and new references (Gao, Ritter, and Zhu, 2013) and (Doidge, Karolyi, and Stultz (2013,2017). We have also removed any (unintended) inference that the innovation of index funds was in response to resolving the problem of too many choices. We thank you for alerting us to this.

4. There are several grammatical issues throughout the study. Below I have provided a non-exhaustive list:

a. On page 2, it should read, “After the introduction, we expect …”

b. On page 2, it should read, “If trading volume in S&P 500 index funds increases over time…”

c. On page 11, it should read, “…by estimating the power law exponent using equation (3)…”

d. On page 11, it should read, “Next, to determine if the effects associated…”

Again, this is a non-exhaustive list. I would recommend that the authors provide a series of careful edits to correct any other grammatical issues.

Response: We thank the reviewer for his/her careful reading. We have made all of these changes as suggested. In addition, we have gone over the manuscript very carefully with an eye to catching any and all stylistic lapses to our exposition. A number of the sentences and paragraphs have now been rewritten. We believe that we have been thorough. Please do, however, let us know if there are any further edits that we may have missed, we are happy to edit and correct errors that might have since crept in. Thank you.

Once again, we thank you for your comments and suggestions. We think that these changes have helped to make the paper stronger and provide more information to a larger set of readers.

---

## [Editor Report · Decision Letter 1]

28 Feb 2020

Evolutionary Disruption of S&P 500 Trading Concentration: An Intriguing Tale of a Financial Innovation

PONE-D-19-27734R1

Dear Dr. Miller,

We are pleased to inform you that your manuscript has been judged scientifically suitable for publication and will be formally accepted for publication once it complies with all outstanding technical requirements.

With kind regards,

Jichang Zhao, Ph.D.

Academic Editor

PLOS ONE
---

## [Editor Report · Acceptance letter]

6 Mar 2020

PONE-D-19-27734R1 

Evolutionary Disruption of S&P 500 Trading Concentration: An Intriguing Tale of a Financial Innovation 

Dear Dr. Miller:

I am pleased to inform you that your manuscript has been deemed suitable for publication in PLOS ONE. Congratulations! Your manuscript is now with our production department. 

With kind regards,

on behalf of

Professor Jichang Zhao 

Academic Editor

PLOS ONE